# Association of an increase in serum albumin levels with positive 1-year outcomes in acute decompensated heart failure: A cohort study

**Takao Kato[1]\*, Hidenori Yaku[1], Takeshi Morimoto[2], Yasutaka Inuzuka[3], Yodo Tamaki[4], Neiko Ozasa[1], Erika Yamamoto[1], Yusuke Yoshikawa[1], Takeshi Kitai[5], Ryoji Taniguchi[6], Moritake Iguchi[7], Masashi Kato[8], Mamoru Takahashi[9], Toshikazu Jinnai[10], Tomoyuki Ikeda[11], Kazuya Nagao[12], Takafumi Kawai[13], Akihiro Komasa[14], Ryusuke Nishikawa[15], Yuichi Kawase[16], Takashi Morinaga[17], Mitsunori Kawato[18], Yuta Seko[19], Masayuki Shiba[1], Mamoru Toyofuku[20], Yutaka Furukawa[5], Kenji Ando[17], Kazushige Kadota[16], Yukihito Sato[6], Koichiro Kuwahara[21], Takeshi Kimura[1]**

1 Department of Cardiovascular Medicine, Kyoto University Graduate School of Medicine, Kyoto, Japan, 2 Department of Clinical Epidemiology, Hyogo College of Medicine, Hyogo, Japan, 3 Department of Cardiovascular Medicine, Shiga General Hospital, Shiga, Japan, 4 Division of Cardiology, Tenri Hospital, Nara, Japan, 5 Department of Cardiovascular Medicine, Kobe City Medical Center General Hospital, Hyogo, Japan, 6 Department of Cardiology, Hyogo Prefectural Amagasaki General Medical Center, Hyogo, Japan, 7 Department of Cardiology, National Hospital Organization Kyoto Medical Center, Kyoto, Japan, 8 Department of Cardiology, Mitsubishi Kyoto Hospital, Kyoto, Japan, 9 Department of Cardiology, Shimabara Hospital, Kyoto, Japan, 10 Department of Cardiology, Japanese Red Cross Otsu Hospital, Shiga, Japan, 11 Department of Cardiology, Hikone Municipal Hospital, Shiga, Japan, 12 Department of Cardiology, Osaka Red Cross Hospital, Osaka, Japan, 13 Department of Cardiology, Kishiwada City Hospital, Osaka, Japan, 14 Department of Cardiology, Kansai Electric Power Hospital, Osaka, Japan, 15 Department of Cardiology, Shizuoka General Hospital, Shizuoka, Japan, 16 Department of Cardiology, Kurashiki Central Hospital, Okayama, Japan, 17 Department of Cardiology, Kokura Memorial Hospital, Fukuoka, Japan, 18 Department of Cardiology, Kobe City Nishi-Kobe Medical Center, Hyogo, Japan, 19 Kitano Hospital, Osaka, Japan, 20 Department of Cardiology, Japanese Red Cross Wakayama Medical Center, Wakayama, Japan, 21 Department of Cardiovascular Medicine, Shinshu University Graduate School of Medicine, Matsumoto, Japan

\* tkato75@kuhp.kyoto-u.ac.jp

**Data Availability Statement:** The minimal data set is ethically restricted by the Institutional Review Board of Kyoto University Hospital. This is because

## Abstract

### Background

Despite the prognostic importance of hypoalbuminemia, the prognostic implication of a change in albumin levels has not been fully investigated during hospitalization in patients with acute decompensated heart failure (ADHF).

### Methods

Using the data from the Kyoto Congestive Heart Failure registry on 3160 patients who were discharged alive for acute heart failure hospitalization and in whom the change in albumin levels was calculated at discharge, we evaluated the association with an increase in serum albumin levels from admission to discharge and clinical outcomes by a multivariable Cox hazard model. The primary outcome measure was a composite of all-cause death or hospitalization for heart failure.

the secondary use of the data was to be reviewed by the Ethics Commission at the time of the initial application. Data are available from the Ethics Committee (contact via TK or directly to ethcom@kuhp.kyoto-u.ac.jp) for researchers who meet the criteria for access to confidential data.

**Funding:** This work was supported by the Japan Agency for Medical Research and Development [18059186] (Drs T. Kato, T. Kuwahara, and N. Ozasa). The founder had no role in the study design, collection, analysis or interpretation of the data, writing the manuscript, or the decision to submit the paper for publication.

**Competing interests:** The authors have declared that no competing interests exist.

## Findings

Patients with increased albumin levels (N = 1083, 34.3%) were younger and less often had smaller body mass index and renal dysfunction than those with no increase in albumin levels (N = 2077, 65.7%). Median follow-up was 475 days with a 96% 1-year follow-up rate. Relative to the group with no increase in albumin levels, the lower risk of the increased albumin group remained significant for the primary outcome measure (hazard ratio: 0.78, 95% confidence interval: 0.69–0.90: P = 0.0004) after adjusting for confounders including baseline albumin levels. When stratified by the quartiles of baseline albumin levels, the favorable effect of increased albumin was more pronounced in the lower quartiles of albumin levels, but without a significant interaction effect (interaction P = 0.49).

## Conclusions

Independent of baseline albumin levels, an increase in albumin during index hospitalization was associated with a lower 1-year risk for a composite of all-cause death and hospitalization in patients with acute heart failure.

## Introduction

Because of the influences of aging, heart failure (HF)-specific disability, and the progression of the disease, improving the long-term mortality in patients with HF hospitalization remains a challenging medical need in developed countries [1, 2]. Hypoalbuminemia is a well-known prognostic marker in patients hospitalized for acute medical illness, including acute decompensated heart failure (ADHF) [3–5]. This prognostic value probably refers primarily to the syndrome of malnutrition-inflammation and the severity of comorbidities [5]. Albumin decreased by the malnutrition, hemodilution, renal loss, and shortened half-life due to severe illness such as inflammation [6]. Serum albumin has many physiological properties, including in particular antioxidant, anti-inflammatory, anticoagulant and anti-platelet aggregation activity. It also plays an essential role in the exchange of fluids across the capillary membrane [6]. In addition to that, hypoalbuminemia may be a potentially modifiable risk factor through the disruption of the protective roles and the exacerbation of peripheral congestion and pulmonary edema [6].

Despite the prognostic importance of hypoalbuminemia, the prognostic implication of a change in albumin levels has not been fully investigated. We aimed to test the hypothesis that an increase as compared with no increase in albumin levels from at admission to at discharge is associated with better 1-year clinical outcome using a large contemporary all-comer registry of patients hospitalized due to ADHF in Japan.

## Materials and methods

### Study population

The Kyoto Congestive Heart Failure registry is a physician-initiated, prospective, observational, multicenter cohort study that enrolled consecutive patients hospitalized for ADHF for the first time between 1 October 2014 and 31 March 2016 without any exclusion criteria [2, 7, 8]. These patients were admitted into 19 secondary and tertiary hospitals, including rural and urban, large and small institutions, throughout Japan. The overall design of the Kyoto

## Patient Flowchart

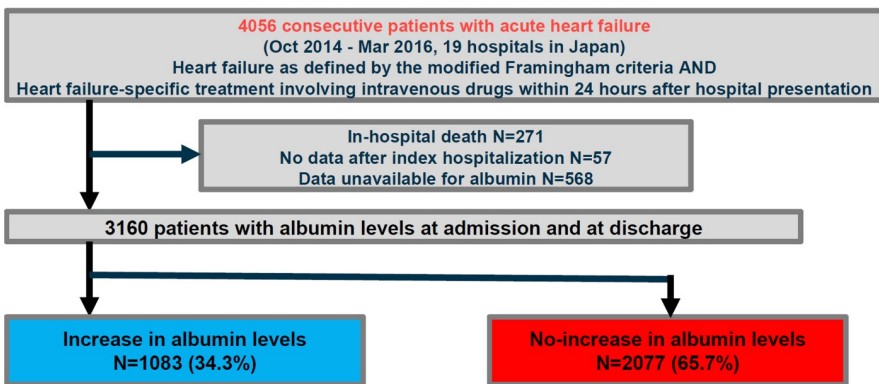

**Fig 1. Study flowchart.** KCHF, Kyoto Congestive Heart Failure.

Congestive Heart Failure study and patient enrolment has been previously described in detail [2, 8, 9]. We enrolled consecutive patients admitted to the participating centers who had ADHF as defined by the modified Framingham criteria and underwent heart failure-specific treatment involving intravenous drugs within 24 hours of hospital presentation. Among 4056 patients enrolled in the registry, we excluded 271 patients who died during index hospitalization, 57 patients who were lost to follow-up, and 568 patients whose albumin levels at admission or at discharge were not available. Therefore, the current study population consisted of 3160 patients who were discharged alive and in whom the change in albumin levels was calculated at discharge (**Fig 1**). We classified the patients into 2 groups: those with increased albumin levels during hospitalization (increase group; N = 1083, 34.3%) and those with no increase in albumin levels (no-increase group; N = 2077, 65.7%) (Fig 1). We compared albumin levels at admission/hospital presentation (anytime of the day) and at the nearest time to discharge in the morning. Any increase in albumin level corresponded to albumin increase. This study is reported in accordance with the Strengthening the Reporting of Observational Studies in Epidemiology (STROBE) reporting guideline. Patient characteristics with missing values for albumin levels were provided in **S1 Table**. Analysis began in October 2019. Detailed definitions of baseline patient characteristics are as follows: 1) Anemia was defined using the World Health Organization criteria (hemoglobin <12.0 g/dL in women and <13.0 g/dL in men). 2) HF was classified according to left ventricular ejection fraction (LVEF) into HF with preserved LVEF (LVEF ≥ 50%), HF with mid-range LVEF (40% ≤ LVEF < 50%), and HF with reduced LVEF (LVEF < 40%) [10].

### Ethics

The investigation conformed with the principles outlined in the Declaration of Helsinki. The study protocol was approved by the ethical committees in Kyoto University Hospital Kyoto University Graduate School of Medicine (approval number: E2311) and participating hospitals (details in **S1 File**). Patient records were anonymized prior to analysis. As the study met the conditions of the Japanese Ethical Guidelines for Medical and Health Research Involving Human Subjects [11], a waiver of written informed consent from each patient was granted by the institutional review boards of Kyoto University and each participating center. We disclosed the details of the present study to the public as an opt-out method, and the notice clearly informed patients of their right to refuse enrollment. Details were described previously [11].

## Outcomes

Clinical follow-up data were collected in October 2017. The attending physicians or research assistants at each participating hospital collected clinical events after index hospitalization from hospital charts or by contacting patients, their relatives, or their referring physicians with consent.

The primary outcome measure for the present analysis was a composite of all-cause death or hospitalization for HF after discharge following index hospitalization. Other outcome measures included all-cause death, cardiovascular death, sudden death, and HF hospitalization. Death was regarded as cardiovascular in origin unless obvious non-cardiovascular causes could be identified. Cardiovascular death included death related to HF, sudden death, death related to stroke, and death from other cardiovascular causes. Sudden death was an unexplained death in a previously stable patient. Stroke was defined as either ischemic or hemorrhagic stroke that required acute or prolonged hospitalization and had symptoms that lasted for more than 24 hours [8]. HF hospitalization was defined as hospitalization due to worsening of HF, requiring intravenous drug therapy [8]. The causes of death were adjudicated by a clinical event committee [12].

## Statistical analyses

We compared the baseline characteristics between the increase and no-increase groups, and we compared 1-year clinical outcomes between the 2 groups and performed subgroup analysis stratified by the albumin levels at admission.

Categorical variables were presented as numbers and percentages, and were compared by the chi square test. Continuous variables are expressed as mean and standard deviation or median with interquartile interval. Based on their distributions, continuous variables were compared using Student's t-test or the Wilcoxon rank-sum test between the 2 groups. The cumulative incidences of clinical events during 1-year after discharge were estimated using the Kaplan-Meier method and the intergroup differences assessed by the log-rank test. Cumulative incidence rates of HF hospitalization or cardiovascular death/ non-cardiovascular death were estimated by using the Gray method [13], accounting for the competing risk of all-cause death or non-cardiovascular death/ cardiovascular death, respectively. We regarded the date of discharge as time zero for clinical follow-up. To estimate the risk for the primary and secondary outcome measures in the increase group relative to the no-increase group, a multivariable Cox proportional-hazards model was developed adjusting for confounders. Consistent with our previous reports [7, 9], we included the following 23 clinically relevant risk-adjusting variables and two additional variables into the model: age ≥80 years, sex, body mass index<22kg/m$^2$, variables related to medical history (previous HF hospitalization, etiology of HF hospitalization associated with acute coronary syndrome, atrial fibrillation or flutter, hypertension, diabetes mellitus, previous myocardial infarction, previous stroke, current smoking, and chronic lung disease), variables related to comorbidities (living alone, ambulatory, systolic blood pressure <90 mmHg, heart rate <60 bpm, LVEF<40%, estimated glomerular filtration rate [eGFR] <30 ml/min/1.73m$^2$, sodium <135 mEq/L, anemia, and albumin levels at admission), and medications at discharge (angiotensin converting enzyme inhibitors or angiotensin II receptor blockers, and β-blockers), along with histories of liver cirrhosis, malignancy, and duration of hospital stay. Multivariable Cox proportional hazards models described by Fine and Gray sub-distribution hazard model [14] were developed for HF hospitalization or cardiovascular death/ non-cardiovascular death accounting for the competing risk of all-cause death or non-cardiovascular death/ cardiovascular death, respectively. Continuous variables were dichotomized by clinically meaningful reference values or median values. Albumin levels at

admission were classified into quartiles: $\leq$ 3.2 g/dl (Q1), > 3.2 g/dl and $\leq$ 3.5 g/dl (Q2), $\geq$3.5 g/dl and $\leq$ 3.8 g/dl (Q3), and > 3.8 g/dl (Q4). The results were expressed as a hazard ratio (HR) and 95% confidence intervals (CIs).

In the sensitivity analysis, we stratified patients into 4 groups based on the quartiles of the percent change of albumin levels as follows: (albumin at discharge) -(albumin at presentation)/(albumin at presentation)*100 (%). Comparisons among 4 groups were performed using the chi-square test for categorical variables and 1-way ANOVA or Kruskal-Wallis test for continuous variables in addition to the Cochran-Armitage trend test in order to assess the trend across the 4 groups. A multivariable Cox proportional-hazards model was developed using the same variables mentioned above to estimate the risk for the primary outcome measures relative to the lowest quartile. In the subgroup analysis, we evaluated the interaction between the albumin levels at admission or the presence of acute coronary syndrome and the risk of the change of the albumin levels for the primary outcome measure. To analyze the baseline factors associated with high LVMI, we used a multivariable logistic regression model involving the following potential independent clinically relevant variable. The adjusted odds ratios and 95% CIs were calculated. All statistical analyses were conducted by a physician (T.K.) and a statistician (T.M.) using JMP 14 and EZR [15]. All the reported P values were two-tailed, and the level of statistical significance was set at P <0.05.

## Results

### Comparison of baseline characteristics between the increase and no-increase groups

Patients with increased albumin levels were younger and more likely to be men and current smokers, less often had a body mass index below 22 kg/m$^2$, eGFR <30 ml/min/1.73m$^2$ and hypertension, and had higher C-reactive protein levels and lower blood urea nitrogen levels and albumin levels at presentation than those with no increase in albumin levels (Table 1). At discharge, those with increased albumin levels had a higher prevalence of diuretics and mineralocorticoid receptor antagonist and had a higher BNP levels than those with no increase in albumin levels.

### Clinical outcomes

The median follow-up duration after discharge was 15.8 months (interquartile interval: 12.1–20.9), with a 96.0% follow-up rate at 1 year. The cumulative 1-year incidence of the primary outcome measure (a composite endpoint of all-cause death or hospitalization for HF) was significantly lower in the increase group than that in the no-increase group (39.6% versus 36.1%, P <0.0001) (Fig 2A). After adjustment for confounders, the lower risk of the increase group relative to the no-increase group remained significant for the primary outcome measure (HR: 0.78, 95% CI: 0.69–0.90, P = 0.0004). For the secondary outcome measures, the cumulative 1-year incidences of all-cause death and hospitalization for HF were significantly lower in the increase group than in the no-increase group (14.4% versus 18.8%, P<0.0001 and 21.2% versus 23.4%, P = 0.0073, respectively) (Fig 2B and 2C). After adjustment for confounders, the lower risk of the increase group relative to the no-increase group remained significant for all-cause death (HR: 0.70, 95% CI: 0.58–0.83, P <0.0001), whereas the lower risk of the increase group relative to the no-increase group was no longer significant for hospitalization for HF (HR: 0.92, 95% CI: 0.72–1.18, P = 0.53) (Table 2 and **S1 Fig**). The trends for cardiovascular and non-cardiovascular death were mostly consistent with that for all-cause death (Table 2).

**Table 1. Patient characteristics.**

| | Increase in albumin (N = 1083, 34.3%) | No-increase in albumin (N = 2077, 65.7%) | P value | N of patients analyzed |
|---|---|---|---|---|
| Clinical characteristics | | | | |
| Age, years | 78 [68–84] | 81 [74–87] | <0.0001 | 3,160 |
| Age >80 years* | 475 (43.9) | 1174 (56.5) | <0.0001 | 3,160 |
| Men* | 623 (57.5) | 1125 (54.2) | 0.08 | 3,160 |
| Body mass index $^{\parallel}$ <22 kg/m$^2$* | 448 (42.8) | 950 (48.4) | 0.003 | 3,009 |
| Prior hospitalization for heart failure* | 373 (34.8) | 747 (36.4) | 0.39 | 3,124 |
| Etiology | | | | |
| Dilated cardiomyopathy | 162 (15.0) | 178 (8.6) | <0.0001 | 3,160 |
| Acute coronary syndrome* | 53 (4.9) | 120 (5.8) | 0.32 | 3,160 |
| Aortic stenosis | 63 (5.8) | 160 (7.7) | 0.057 | 3,160 |
| Hypertensive | 255 (23.6) | 538 (25.9) | 0.15 | 3,160 |
| Ischemic (not acute) | 278 (25.7) | 580 (27.9) | 0.18 | 3,160 |
| Others | 272 (25.1) | 501 (24.1) | 0.54 | 3,160 |
| Medical history | | | | |
| Atrial fibrillation or flutter* | 444 (41.0) | 874 (42.1) | 0.57 | 3,160 |
| Hypertension* | 747 (69.0) | 1564 (75.3) | 0.0002 | 3,160 |
| Diabetes mellitus* | 408 (37.7) | 788 (37.9) | 0.91 | 3,160 |
| Dyslipidemia | 445 (41.1) | 809 (39.0) | 0.25 | 3,160 |
| Prior myocardial infarction* | 244 (22.5) | 482 (23.2) | 0.69 | 3,160 |
| Prior stroke* | 180 (16.6) | 332 (16.0) | 0.65 | 3,160 |
| Current smoking* | 165 (15.4) | 215 (10.6) | 0.0001 | 3,104 |
| Ventricular tachycardia/fibrillation | 54 (5.0) | 71 (3.4) | 0.03 | 3,160 |
| Chronic lung disease* | 136 (12.6) | 291 (14.0) | 0.27 | 3,160 |
| Liver cirrhosis* | 12 (1.1) | 28 (1.4) | 0.62 | 3,160 |
| Malignancy* | 133 (12.3) | 328 (15.8) | 0.008 | 3,160 |
| Social backgrounds | | | | |
| Poor medical adherence | 208 (19.2) | 339 (16.3) | 0.048 | 3,160 |
| Living alone* | 244 (22.5) | 422 (20.3) | 0.15 | 3,160 |
| Daily life activities | | | | |
| Ambulatory* | 861 (80.1) | 1627 (79.1) | 0.55 | 3,132 |
| Use of wheelchair [outdoor only] | 83 (7.7) | 159 (7.7) | 1.00 | 3,132 |
| Use of wheelchair [outdoor and indoor] | 97 (9.0) | 193 (9.4) | 0.80 | 3,132 |
| Bedridden | 34 (3.2) | 78 (3.8) | 0.42 | 3,132 |
| Vital signs at presentation | | | | |
| Systolic blood pressure, mmHg | 145 ± 33 | 150 ± 36 | 0.0001 | 3,149 |
| Systolic blood pressure <90 mmHg* | 33 (3.1) | 45 (2.2) | 0.15 | 3,153 |
| Heart rate, bpm | 97 ± 29 | 96 ± 27 | 0.45 | 3,142 |
| Heart rate <60 bpm* | 78 (7.2) | 137 (6.6) | 0.55 | 3,142 |
| Body temperature >37.5 degree Celsius | 55 (5.3) | 141 (7.1) | 0.06 | 3,033 |
| NYHA Class III or IV | 917 (85.0) | 1838 (88.9) | 0.0021 | 3,147 |
| Tests at presentation | | | | |
| LVEF | 46.0 ± 16.4 | 46.6 ± 16.1 | 0.35 | 3,086 |
| HFrEF (LVEF <40%)* | 397 (36.7) | 761 (36.8) | 1.00 | 3,153 |
| HFmrEF (LVEF 40–49%) | 240 (22.2) | 372 (18.0) | 0.005 | 3,153 |
| HFpEF (LVEF ≥50%) | 445 (41.1) | 938 (45.3) | 0.026 | 3,153 |
| BNP, ng/L | 691 [378–1245] | 724 [395–1268] | 0.21 | 2,811 |
| Serum creatinine, mg/dL | 1.05 [0.80–1.50] | 1.12 [0.83–1.67] | 0.0006 | 3,160 |

(*Continued*)

**Table 1.** (Continued)

| | Increase in albumin (N = 1083, 34.3%) | No-increase in albumin (N = 2077, 65.7%) | P value | N of patients analyzed |
|---|---|---|---|---|
| eGFR <30 mL/min/1.73m$^{2*}$ | 241 (22.3) | 604 (29.1) | <0.0001 | 3,160 |
| Blood urea nitrogen, mg/dL | 22 [16–32] | 25 [18–37] | <0.0001 | 3,157 |
| Sodium <135 mEq/L* | 126 (11.7) | 255 (12.3) | 0.61 | 3,155 |
| Anemia*§ | 740 (68.4) | 1374 (66.2) | 0.23 | 3,157 |
| C reactive protein, mg/dL | 0.69 [0.22–2.57] | 0.60 [0.20–1.85] | 0.0001 | 3,103 |
| Increase in albumin levels, g/dl† | | | | |
| Albumin levels at presentation, g/dl *# | 3.25 ± 0.48 | 3.58 ± 0.45 | <0.0001 | 3,160 |
| Q1: ≤ 3.2 g/dl | 524 (48.4) | 463 (22.3) | <0.0001 | 987 |
| Q2: > 3.2 g/dl and ≤ 3.5 g/dl | 262 (24.2) | 451 (21.7) | 0.17 | 713 |
| Q3: ≥3.5 g/dl and ≤ 3.8 g/dl | 193 (17.8) | 555 (26.7) | <0.0001 | 748 |
| Q4: > 3.8 g/dl | 104 (9.6) | 608 (29.3) | <0.0001 | 712 |
| Medications at discharge | | | | |
| ACE-I or ARB* | 634 (58.5) | 1177 (56.7) | 0.31 | 3,160 |
| Beta blocker* | 740 (68.3) | 1358 (65.4) | 0.10 | 3,160 |
| MRA | 520 (48.0) | 898 (43.2) | 0.010 | 3,160 |
| Diuretics (non-MRA) | 945 (87.2) | 1694 (81.6) | <0.0001 | 3,160 |
| BNP at discharge, ng/L | 231 [118–475] | 299 [147–543] | <0.0001 | 2,148 |
| Hospital stay >16 days* (median) | 557 (51.4%) | 979 (47.1%) | 0.02 | 3,160 |

*26 risk-adjusting variables selected for COX hazard model including albumin quartiles.

|| Body mass index was calculated as weight in kilograms divided by height in meters squared.

§ Anemia was defined by the World Health Organization criteria (hemoglobin <12.0 g/dL in women and <13.0 g/dL in men).

† Any increase in albumin level corresponded to albumin increase.

#Albumin levels at admission were classified into quartiles: ≤ 3.2 g/dl (Q1), > 3.2 g/dl and ≤ 3.5 g/dl (Q2), ≥3.5 g/dl and ≤ 3.8 g/dl (Q3), and > 3.8 g/dl (Q4).

BP = blood pressure; bpm = beat per minute; NYHA = New York Heart Association, LVEF = left ventricular ejection fraction; HFrEF = heart failure with reduced ejection fraction; HFmrEF = heart failure with mid-range ejection fraction; HFpEF = heart failure with preserved ejection fraction; BNP = brain-type natriuretic peptide; eGFR = estimated glomerular filtration rate, ACE-I = angiotensin converting enzyme inhibitors, ARB = angiotensin II receptor blockers, MRA = mineralocorticoid receptor antagonist.

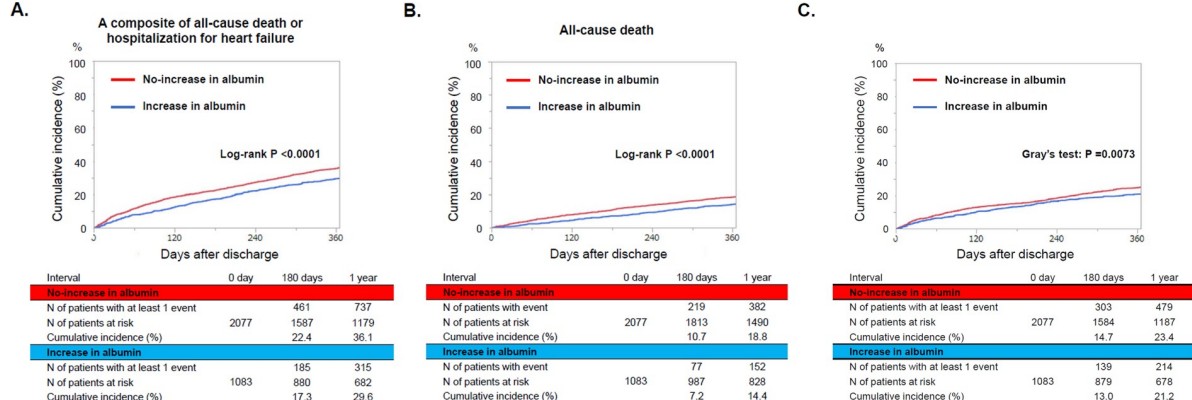

**Fig 2.** Kaplan Meier curves for (A) the primary outcome measure, (B) all-cause death, and (C) hospitalization for heart failure. The primary outcome measure was defined as a composite of all-cause death or hospitalization for heart failure.

**Table 2. Primary and secondary outcomes.**

| | Increase in albumin | No-increase in albumin | Unadjusted Hazard Ratio (95%CI) | P value | Adjusted Hazard Ratio (95%CI) | P value |
|---|---|---|---|---|---|---|
| | N of patients with event/N of patients at risk (Cumulative 1-year incidence) | N of patients with event/N of patients at risk (Cumulative 1-year incidence) | | | | |
| A composite of all-cause death or HF hospitalization | 382/1083 (29.6%) | 887/2077 (36.1%) | 0.77 (0.68–0.87) | <0.0001 | 0.78 (0.68–0.89) | 0.0004 |
| All-cause death | 204/1083 (14.4%) | 520/2077 (18.8%) | 0.71 (0.60–0.83) | <0.0001 | 0.69 (0.58–0.83) | <0.0001 |
| HF Hospitalization | 247/1083 (21.2%) | 545/2077 (23.4%) | 0.85 (0.73–0.98) | 0.03 | 0.92 (0.72–1.18) | 0.53 |
| Cardiovascular death | 125/1083 (8.8%) | 307/2077 (13.0%) | 0.76 (0.61–0.93) | 0.008 | 0.81 (0.64–1.04) | 0.09 |
| Non-cardiovascular death | 79/1083 (5.5%) | 213/2077 (7.6%) | 0.69 (0.53–0.89) | 0.0048 | 0.62 (0.46–0.82) | 0.001 |

HF = heart failure, CI = confidence interval, HR = hazard ratio.

## Sensitivity analysis based on the percent change of albumin levels

We stratified patients into 4 groups based on the quartiles of the percent change of albumin levels: the lowest quartile ($\leq$ -11.1%), the lower quartile ($>$ -11.1% and $\leq$ -3.0%), the higher quartile ($>$ -3.0% and $\leq$ 5.3%), and the highest quartile ($>$ 5.3%) (**S2 Table**). The cumulative 1-year incidence of the primary outcome measure (a composite endpoint of all-cause death or hospitalization for HF) was significantly lower in the higher and the highest quartiles than that in the lowest quartile (**S2 Fig**). The lower risk of the higher and the highest percent change groups relative to the lowest percent change group was significant after adjustment (HR: 0.70, 95% CI: 0.58–0.83, P <0.0001) (**S3 Table**).

## Subgroup analysis based on the baseline albumin levels

When stratified by the quartiles of baseline albumin levels, the lower risk of the increase group relative to the no-increase group for the primary outcomes and all-cause death trended to be more prominent in the lower quartiles of albumin levels, but without significant interaction across the quartiles (interaction P = 0.49 for the primary outcomes, and interaction P = 0.89 for all-cause death) (Table 3).

## Subgroup analysis based on the acute coronary syndrome

When stratified by the acute coronary syndrome, the lower risk of the increase group relative to the no-increase group for the primary outcomes and all-cause death was consistently seen in patients with or without acute coronary syndrome. The effect was directionally significant in patients with acute coronary syndrome (**S4 Table**).

## Baseline factors associated with the increase in albumin levels

According to the multivariable logistic regression analysis, the albumin levels at admission < 3 g/dL and anemia were independent positive factors associated with the increase in albumin levels, while age $\geq$80 years, eGFR<30 ml/min/1.73m$^2$, body mass index<22kg/m$^2$, NYHA III/IV at presentation, and the history of diabetes mellitus were independent negative factors associated with the increase in albumin levels (**Table 4**).

## Discussions

The principal findings of the present study were as follows: 1) The increase as compared with no increase in albumin level during the index hospitalization for ADHF was associated with

**Table 3. Subgroup analysis based on the baseline albumin levels.**

| | | Increase in albumin | No-increase in albumin | Unadjusted | | Adjusted | | |
|---|---|---|---|---|---|---|---|---|
| | | N of patients with events/N of patients at risk (Cumulative 1-year incidence) | N of patients with events/N of patients at risk (Cumulative 1-year incidence) | HR (95% CI) | P value | HR (95% CI) | P value | P value for interaction |
| A composite of death or HF hospitalization | Q1 | 213/524 (33.9%) | 251/463 (47.7%) | 0.63 (0.52–0.76) | <0.0001 | 0.75 (0.62–0.92) | 0.006 | 0.49 |
| | Q2 | 94/262 (29.7%) | 224/451 (41.7%) | 0.65 (0.51–0.82) | 0.0003 | 0.79 (0.61–1.03) | 0.08 | |
| | Q3 | 43/193 (18.4%) | 250/555 (32.0%) | 0.54 (0.39–0.75) | 0.0001 | 0.79 (0.55–1.13) | 0.19 | |
| | Q4 | 32/104 (28.5%) | 207/608 (26.9%) | 0.94 (0.63–1.34) | 0.73 | 1.01 (0.64–1.52) | 0.98 | |
| All-cause death | Q1 | 128/524 (18.1%) | 176/463 (30.3%) | 0.55 (0.44–0.70) | <0.0001 | 0.72 (0.55–0.92) | 0.01 | 0.89 |
| | Q2 | 48/262 (14.5%) | 143/451 (23.7%) | 0.53 (0.38–0.72) | <0.0001 | 0.65 (0.44–0.93) | 0.02 | |
| | Q3 | 16/193 (6.3%) | 115/555 (14.9%) | 0.36 (0.21–0.59) | <0.0001 | 0.64 (0.37–1.12) | 0.12 | |
| | Q4 | 12/104 (10.0%) | 86/608 (10.1%) | 0.85 (0.44–1.48) | 0.58 | 0.93 (0.44–1.83) | 0.85 | |

Albumin levels at admission were classified into quartiles: ≤ 3.2 g/dl (Q1), > 3.2 g/dl and ≤ 3.5 g/dl (Q2), ≥3.5 g/dl and ≤ 3.8 g/dl (Q3), and > 3.8 g/dl (Q4).

CI = confidence interval, HR = hazard ratio.

lower adjusted risk for the primary outcome measure (a composite of all-cause death or hospitalization for HF) as well as all-cause death, cardiovascular death, and non-cardiovascular death. 2) There was no interaction between the albumin levels at admission and the effects of the increase relative to no increase in albumin for the primary outcome measure and all-cause death.

In patients with ADHF, albumin levels are influenced by the production from the liver, hemodilution, loss from the kidney, digestive tract, and vascular bed, and shortened half-life due to severe illness or inflammation [3, 4, 16–18]. Comparisons of albumin levels between the decompensated state at admission and the compensated condition at discharge may enable assessment of the recovery of multi-organ damage due to the worsening of HF. In this study, about two-thirds of patients did not show increased albumin levels at discharge. They may be considered in the process of recovering from ADHF when their albumin levels were low. Nakayama et al reported that an increase in serum albumin during hospitalization had favorable long-term prognostic impact on the composite endpoint of all-cause death or HF hospitalization in 115 patients from a single center [19]. In the present study, we also showed a lower risk for all-cause death in the group with increased albumin levels during hospitalization after adjustment of baseline albumin levels. This impact seemed to be greater in patients with low serum albumin levels at admission, although the interaction was not significant. In addition, CRP levels at admission was significantly higher in the group with increased albumin levels at discharge. This finding implied that, patients with the increase in albumin levels showed the

**Table 4. Factors associated with the increase in albumin levels.**

| | Adjusted OR | Lower 95% CI | Upper 95% CI | P value |
|---|---|---|---|---|
| Albumin <3 mg/dl | 3.94 | 3.14 | 4.95 | <0.0001 |
| Age ≥80 years | 0.54 | 0.46 | 0.65 | <0.0001 |
| eGFR<30 mL/min/1.73m2 | 0.65 | 0.53 | 0.80 | <0.0001 |
| BMI<22 kg/m$^2$ | 0.80 | 0.67 | 0.94 | 0.0093 |
| Anemia | 1.26 | 1.05 | 1.51 | 0.01 |
| NYHA III/IV | 0.76 | 0.60 | 0.96 | 0.02 |
| DM | 0.83 | 0.70 | 0.997 | 0.046 |
| COLD | 0.84 | 0.66 | 1.07 | 0.16 |
| SBP<90 mmHg | 1.42 | 0.85 | 2.36 | 0.17 |
| History of stroke | 1.12 | 0.90 | 1.39 | 0.30 |
| HFrEF | 0.92 | 0.76 | 1.10 | 0.36 |
| AF | 1.04 | 0.88 | 1.23 | 0.61 |
| History of MI | 1.02 | 0.84 | 1.24 | 0.81 |
| BNP or NT-pro BNP above median* | 1.01 | 0.86 | 1.20 | 0.82 |
| History of HF hospitalization | 1.01 | 0.85 | 1.21 | 0.84 |
| Female | 1.01 | 0.85 | 1.20 | 0.85 |

*BNP value above 715.9 ng/L or NT-pro-BNP> 5744.1 pg/L.

OR = odds ratio, CI = confidence interval, eGFR = estimated glomerular filtration rate; DM = diabetes mellitus; BMI = body mass index; HFrEF = HF with reduced EF (<40%); COLD = chronic obstructive lung disease, SBP = systolic blood pressure; AF = atrial fibrillation/flutter; ACS = acute coronary syndrome; NYHA = New York Heart Association.

decreased albumin levels at admission due to the serious conditions; however, once recovered, they showed the less risk for mortality than those without. Factors positively associated with the increase in albumin levels were low baseline albumin levels and anemia in the present study. Due to the nature of the observational studies, a causal relationship could not be determined in this study. However, an attempt to increase the albumin levels through the improvement in nutritional status in patients with HF is under investigation [20, 21]. Whether nutritional support promotes the recovery of multi-organ damage due to worsening of HF needs to be determined by clinical trials, and dietary recommendations should be provided at discharge to patients hospitalized for ADHF [22]. In addition, patients with no increase in albumin levels had a higher BNP levels at discharge, although the difference was not large. From a clinical point of view, the assessment of intravascular volume status is crucial when we interpret the change of albumin levels.

We could not assess the intra-individual variability. In order not only to evaluate the prognostic impact of any increase in albumin but also to evaluate the increase that is above the intra-individual variability, we added the sensitivity analysis on the percent change of albumin levels. From a prognostic point of view, the cut-off point might exist around -3.0%; however, this value may be validated by further studies. In addition, whether the increase of albumin can be associated with a lower risk for HF hospitalization should be determined by further studies.

## Limitations

Several limitations of the present study should be noted. First, we did not analyze nor did we have the data for the serial measurements of albumin, and thus only compared the values at admission and at discharge. Second, the importance of the increase in albumin levels in patients with normal albumin levels needs further investigation, although 47.9% of patients

with ADHF showed decreased albumin levels [5]. We included the baseline albumin values in the risk-adjusting variables and also performed subgroup analysis based on the baseline albumin values, showing a lower risk in the group with increased albumin levels during hospitalization regardless of the baseline albumin levels. Third, there may be unadjusted confounding factors present in the current study. Finally, those excluded for missing data included patients without hypertension and anemia.

## Conclusion

Independent of baseline albumin levels, an increase of albumin was associated with a lower 1-year risk for all-cause death in patients hospitalized for ADHF.

## Supporting information

**S1 Checklist. STROBE statement—checklist of items that should be included in reports of *cohort studies*.**
(DOCX)

**S1 File. Ethical approval of other participating centers.**
(DOCX)

**S1 Fig. Adjusted survival curves.** (A) A composite of all cause death and hospitalization for heart failure. (B) All-cause death. (C) Heart failure hospitalization. We adjusted outcomes for the same variables in COX hazard model.
(DOCX)

**S2 Fig. Kaplan Meier curves for the primary outcome measure stratified by quartiles of the percent change of albumin levels.**
(DOCX)

**S1 Table. Baseline characteristics of the patients with versus without albumin data.**
(DOCX)

**S2 Table. Patient characteristics based on the quartiles of the percent change of albumin levels.**
(DOCX)

**S3 Table. Outcomes based on the quartiles of the percent change of albumin levels.**
(DOCX)

**S4 Table. Subgroup analysis with or without acute coronary syndrome.**
(DOCX)

## Author Contributions

**Conceptualization:** Takao Kato, Yasutaka Inuzuka, Yodo Tamaki, Neiko Ozasa, Erika Yamamoto, Mamoru Takahashi, Yuta Seko, Yukihito Sato.

**Data curation:** Takao Kato, Hidenori Yaku, Yasutaka Inuzuka, Yodo Tamaki, Neiko Ozasa, Erika Yamamoto, Yusuke Yoshikawa, Takeshi Kitai, Ryoji Taniguchi, Moritake Iguchi, Masashi Kato, Toshikazu Jinnai, Tomoyuki Ikeda, Takafumi Kawai, Akihiro Komasa, Ryusuke Nishikawa, Yuichi Kawase, Takashi Morinaga, Mitsunori Kawato, Yuta Seko, Masayuki Shiba, Mamoru Toyofuku, Yutaka Furukawa, Kenji Ando, Kazushige Kadota, Yukihito Sato, Koichiro Kuwahara.

**Formal analysis:** Takao Kato, Takeshi Morimoto.

**Investigation:** Takao Kato, Hidenori Yaku, Yasutaka Inuzuka, Yodo Tamaki, Neiko Ozasa, Erika Yamamoto, Yusuke Yoshikawa, Takeshi Kitai, Ryoji Taniguchi, Moritake Iguchi, Masashi Kato, Mamoru Takahashi, Toshikazu Jinnai, Tomoyuki Ikeda, Kazuya Nagao, Takafumi Kawai, Akihiro Komasa, Ryusuke Nishikawa, Yuichi Kawase, Takashi Morinaga, Mitsunori Kawato, Yuta Seko, Masayuki Shiba, Mamoru Toyofuku, Yutaka Furukawa, Kenji Ando, Kazushige Kadota, Yukihito Sato, Koichiro Kuwahara.

**Methodology:** Takao Kato, Hidenori Yaku, Takeshi Morimoto, Yasutaka Inuzuka, Yodo Tamaki, Neiko Ozasa, Erika Yamamoto, Yusuke Yoshikawa, Takeshi Kitai, Ryoji Taniguchi, Moritake Iguchi, Masashi Kato, Mamoru Takahashi, Toshikazu Jinnai, Tomoyuki Ikeda, Kazuya Nagao, Takafumi Kawai, Akihiro Komasa, Ryusuke Nishikawa, Yuichi Kawase, Takashi Morinaga, Mitsunori Kawato, Yuta Seko, Masayuki Shiba, Mamoru Toyofuku, Yutaka Furukawa, Kenji Ando, Kazushige Kadota, Yukihito Sato, Koichiro Kuwahara, Takeshi Kimura.

**Project administration:** Takao Kato, Hidenori Yaku, Takeshi Morimoto, Yasutaka Inuzuka, Yodo Tamaki, Neiko Ozasa, Erika Yamamoto, Takeshi Kimura.

**Supervision:** Takeshi Kimura.

**Writing – original draft:** Takao Kato.

**Writing – review & editing:** Hidenori Yaku, Takeshi Morimoto, Yasutaka Inuzuka, Yodo Tamaki, Neiko Ozasa, Erika Yamamoto, Yusuke Yoshikawa, Takeshi Kitai, Ryoji Taniguchi, Moritake Iguchi, Masashi Kato, Mamoru Takahashi, Toshikazu Jinnai, Tomoyuki Ikeda, Kazuya Nagao, Takafumi Kawai, Akihiro Komasa, Ryusuke Nishikawa, Yuichi Kawase, Takashi Morinaga, Mitsunori Kawato, Yuta Seko, Masayuki Shiba, Mamoru Toyofuku, Yutaka Furukawa, Kenji Ando, Kazushige Kadota, Yukihito Sato, Koichiro Kuwahara, Takeshi Kimura.

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
