## [Decision Letter · Decision Letter 0]

19 Oct 2020

PONE-D-20-30330

Association of an increase in serum albumin levels with positive 1-year outcomes in acute decompensated heart failure: A cohort study

PLOS ONE

Dear Dr. Kato,

Thank you for submitting your manuscript to PLOS ONE. After careful consideration, we feel that it has merit but does not fully meet PLOS ONE’s publication criteria as it currently stands. Therefore, we invite you to submit a revised version of the manuscript that addresses the points raised during the review process.

 All issues raised by expert reviewers are required.

We look forward to receiving your revised manuscript.

Kind regards,

Vincenzo Lionetti, M.D., PhD

Academic Editor

PLOS ONE

Journal Requirements:

2. In your Methods section, please provide additional information about the statistical analysis performed, for example by describing any correction for multiple comparisons performed.

Reviewers' comments:

Reviewer's Responses to Questions

**Comments to the Author**

1. Is the manuscript technically sound, and do the data support the conclusions?

Reviewer #1: Yes

Reviewer #2: Yes

2. Has the statistical analysis been performed appropriately and rigorously? 

Reviewer #1: Yes

Reviewer #2: Yes

3. Have the authors made all data underlying the findings in their manuscript fully available?

Reviewer #1: Yes

Reviewer #2: Yes

4. Is the manuscript presented in an intelligible fashion and written in standard English?

Reviewer #1: Yes

Reviewer #2: Yes

5. Review Comments to the Author

Reviewer #1: The present paper is an analysis of the Kyoto Congestive Heart Failure registry evaluating 3160 patients who were 1) discharged after an hospitalization for acute heart failure (HF) and 2) had serum albumin dosed at admission and at discharge. The Authors evaluated the relationship between any increase in serum albumin and all-cause death or HF hospitalization over a median 475 day follow-up. Patients with increased albumin (34%) had a lower risk of this outcome, even after adjusting for baseline albumin. The prognostic impact of increased albumin seemed greater in patients in the lower quartiles of baseline albumin. While there are no major flaws in the analysis and the results are reasonable, I have some doubts about the evaluation of the prognostic impact of "any increase in albumin", instead of an increase that is above the intra-individual variability of this biomarker. Please find below some other comments.

Abstract: "An intervention to increase the albumin levels in the treatment for ADHF needs to be investigated". I would delete this statement in the Abstract.

Introduction: please expand on the causes and prognostic impact of hypoalbuminemia.

Methods:

- "Death was regarded as cardiovascular in origin unless obvious noncardiovascular

causes could be identified". I would prefer to adjudicate as cardiovascular deaths only the cases with clear evidence of cardiovascular disease as the underlying cause, non-cardiovascular deaths those where there were other obvious cases, and classify as "unclear" the cause of the death in the remaining cases.

- Your definitions of "chronic kidney disease" and particularly "renal dysfunction" are questionable.

- "Interquartile range" would be better replaced by "interquartile interval".

- Page 15: "HF hospitalization associated with acute coronary syndrome". ACS events, possibly complicated by HF, should be clearly differentiated from acute HF.

Results:

- Table 1: BNP would be better expressed as ng/L.

- The percent changes in albumin in the 2 groups (any increase vs. no increase) should be reported. You may consider stratifying patients based on certain thresholds of delta % albumin.

- Follow-up duration could be converted in months.

- In your survival analysis, you should take into account competing risks (cardiovascular death vs. non-cardiovascular death, HF hospitalization vs. all-cause death).

- Splitting the population into two broad categories (any increase vs. no increase) instead of evaluating delta % changes in albumin results in a loss of information (doi: 10.1002/sim.2331). You should consider evaluating the prognostic value of delta % changes, possibly adjusting for disease severity, the duration of hospital admission, and other potential confounders.

Reviewer #2: In this paper, Kato et al. aimed at assessing the prognostic implication of a change in albumin during hospitalization for acute decompensated heart failure (ADHF) on prognosis.

The data of this study are derived from the Kyoto Congestive Heart Failure registry, a physician-initiated, prospective, observational, multicenter cohort study that enrolled consecutive patients hospitalized for ADHF October 2014 and March 2016.

This allowed the inclusion of an important number of patients (3160 patients).

The study is well written, and the results are interesting.

Nevertheless, I have some concerns that need to be addressed.

1) The authors state that patients were divided into 2 groups according to the increase or the absence of increase in albumin levels before discharge.

How the increase in albumin level was defined. A simple increase in 1 unit of albumin was sufficient to define increase, or a stricter definition was used?

2) The paragraph on Ethics is too long. For the publication, I think it is necessary to indicate that “The investigation conformed with the principles outlined in the Declaration of Helsinki and approved by the ethical committees of the participating hospitals”. A detailed list of these hospitals and corresponding approbation numbers should be provided as supplementary material.

3) The paragraph “Definition” doesn’t identify the content of the paragraph itself. I suggest indicating the title of this paragraph as “Outcome”.

4) The “detailed definition of patients characteristics should be inserted in the paragraph “Population”.

5) The authors should add some data about albumin assessment. The fact that they consider variation in albumin concentration during hospitalization is indicated in the introduction, but this should be clearly indicated in the method. They should indicate when Albumine was assessed at admission and before discharge. Any increase in albumin level (even the smallest one) corresponded to albumin increase? This should be defined in detail.

6) Can the author indicate the cause of hospital admission?

7) In table 1, the p-value for HFmEF and HFpEF is not indicated and should be added

8) In table 1, the definition “Albumin, g/dl *☨” should be replaced by “Increase in albumin levels”.

9) In table 1, the p-value for the different quartiles of albumin is not indicated

10) Were patients taking also non-MRA diuretics at discharge?

11) Table 2 is difficult to read. I think the authors should simply indicate unadjusted an unadjusted HR in this table and maybe add the comparison of events among groups in Table 1. A similar comment can be applied to Table 3.

12) From a clinical point of view, were all patients discharged in a condition of euvolemia? Did the study include patients admitted for cardiogenic shock?

13) In the conclusion the authors state that “Independent of baseline albumin levels, an increase of albumin was associated with a lower 1-year risk for the composite of all-cause death or HF hospitalization in patients hospitalized for ADHF”. P-value for the adjusted risk of HF is >0.05, which means that an increase in albumin level is not associated with a significant reduction in HF hospitalization. The statement should be corrected and these results should put in perspective in the discussion.

14) I think that all the survival curves should indicate the adjusted comparison of survival/hospitalization curve and not unadjusted data. Otherwise, the results are largely misleading.

15) Were the authors able to identify parameters associated with the increase in albumin levels during hospitalization?

6. PLOS authors have the option to publish the peer review history of their article (what does this mean?). If published, this will include your full peer review and any attached files.

Reviewer #1: No

Reviewer #2: No

---

## [Author Response · Author response to Decision Letter 0]

23 Nov 2020

We thank the reviewers for careful assessment and positive comments.

Reviewer #1: The present paper is an analysis of the Kyoto Congestive Heart Failure registry evaluating 3160 patients who were 1) discharged after an hospitalization for acute heart failure (HF) and 2) had serum albumin dosed at admission and at discharge. The Authors evaluated the relationship between any increase in serum albumin and all-cause death or HF hospitalization over a median 475 day follow-up. Patients with increased albumin (34%) had a lower risk of this outcome, even after adjusting for baseline albumin. The prognostic impact of increased albumin seemed greater in patients in the lower quartiles of baseline albumin. While there are no major flaws in the analysis and the results are reasonable, I have some doubts about the evaluation of the prognostic impact of "any increase in albumin", instead of an increase that is above the intra-individual variability of this biomarker. 

Response: 

We thank the reviewer for very important comments. In order to evaluate the prognostic impact of "any increase in albumin", instead of an increase that is above the intra-individual variability, we also included the sensitivity analysis on the percent change of albumin levels. We assessed the outcomes according to the quartiles of the % change and showed that the higher and the highest quartiles were significantly associated with the lower risk relative to the lowest quartile of the % change. 

“We compared albumin levels at admission/hospital presentation (anytime of the day) and at the nearest time to discharge in the morning. Any increase in albumin level corresponded to albumin increase.” (Methods, page 5, lines 19-21)

“Sensitivity analysis based on the percent change of albumin levels

We stratified patients into 4 groups based on the quartiles of the percent change of albumin levels: the lowest quartile (≤ -11.1%), the lower quartile (> -11.1% and ≤ -3.0%), the higher quartile (> -3.0% and ≤ 5.3%), and the highest quartile (> 5.3%) (S4 Table). The cumulative 1-year incidence of the primary outcome measure (a composite endpoint of all-cause death or hospitalization for HF) was significantly lower in the higher and the highest quartiles than that in the lowest quartile (S5 Figure). The lower risk of the higher and the highest percent change groups relative to the lowest percent change group was significant after adjustment (HR: 0.70, 95% CI: 0.58-0.83, P <0.0001) (S6 Table)”. (Results, page 16)

“We could not assess the intra-individual variability. In order not only to evaluate the prognostic impact of any increase in albumin but also to evaluate the increase that is above the intra-individual variability, we added the sensitivity analysis on the percent change of albumin levels. From a prognostic point of view, the cut-off point might exist around -3.0%; however, this value may be validated by further studies”. (Discussion, page 21, lines 12-16)

Please find below some other comments.

Abstract: "An intervention to increase the albumin levels in the treatment for ADHF needs to be investigated". I would delete this statement in the Abstract.

Response:

Thank you for your suggestions. I have removed this sentence in the Abstract.

Introduction: please expand on the causes and prognostic impact of hypoalbuminemia.

Response: 

We appreciate your comments. Now we have expanded the introduction section as follows: 

“This prognostic value probably refers primarily to the syndrome of malnutrition-inflammation and the severity of comorbidities (5). Albumin decreased by the malnutrition, hemodilution, renal loss, and shortened half-life due to severe illness such as inflammation (6). Serum albumin has many physiological properties, including in particular antioxidant, anti-inflammatory, anticoagulant and anti-platelet aggregation activity. It also plays an essential role in the exchange of fluids across the capillary membrane (6). In addition to that, hypoalbuminemia may be a potentially modifiable risk factor through the disruption of the protective roles and the exacerbation of peripheral congestion and pulmonary edema (6).” (Introduction, page 4, lines 6-14)

Methods:

- "Death was regarded as cardiovascular in origin unless obvious noncardiovascular

causes could be identified". I would prefer to adjudicate as cardiovascular deaths only the cases with clear evidence of cardiovascular disease as the underlying cause, non-cardiovascular deaths those where there were other obvious cases, and classify as "unclear" the cause of the death in the remaining cases.

Response:

We thank the reviewer for the important points. The definition of the death in the present study was in accordance with the previous studies in this registry. The clear identification of death is actually difficult when the patients with heart failure often accompany the infection such as pneumonia, although we rigorously collected the data; thus, we adopted the all cause death (and heart failure hospitalization) as the primary outcome in the main analysis.

- Your definitions of "chronic kidney disease" and particularly "renal dysfunction" are questionable.

Response: 

We appreciate your comments. We deleted the unclear or questionable definitions and use the term eGFR<30 ml/min/1.73m2.

- "Interquartile range" would be better replaced by "interquartile interval".

Response:

We have replaced interquartile range by interquartile interval.

- Page 15: "HF hospitalization associated with acute coronary syndrome". ACS events, possibly complicated by HF, should be clearly differentiated from acute HF.

Response:

We thank the reviewer for the careful assessment. We included the patients hospitalized with acute decompensated heart failure with any cause. As the reviewer pointed out, ACS is one of the causes of acute decompensation and should be differentiated. Thus, we have added the subgroup analysis stratified by ACS.

“Subgroup analysis based on the acute coronary syndrome

When stratified by the acute coronary syndrome, the lower risk of the increase group relative to the no-increase group for the primary outcomes and all-cause death was consistently seen in patients with or without acute coronary syndrome. The effect was directionally significant in patients with acute coronary syndrome (S7 Table).” (Result, page 18)

Results:

- Table 1: BNP would be better expressed as ng/L. 

Response: 

We have changed the expression.

- The percent changes in albumin in the 2 groups (any increase vs. no increase) should be reported. You may consider stratifying patients based on certain thresholds of delta % albumin.

Response:

We also included the % change in the 2 groups. We also included the analysis stratified by the quartile based on the % change as mentioned above.

- Follow-up duration could be converted in months.

Response:

We have converted the follow-up duration in months.

- In your survival analysis, you should take into account competing risks (cardiovascular death vs. non-cardiovascular death, HF hospitalization vs. all-cause death).

Response:

We have modified the survival analysis with competing risks regarding CV death vs. non-CV death and HF hospitalization vs. all-cause death. We have added the Fine and Grays’ method in the method section.

- Splitting the population into two broad categories (any increase vs. no increase) instead of evaluating delta % changes in albumin results in a loss of information (doi: 10.1002/sim.2331). You should consider evaluating the prognostic value of delta % changes, possibly adjusting for disease severity, the duration of hospital admission, and other potential confounders.

Response:

We thank the reviewer for very important comments. As a sensitivity analysis, we included the % change of albumin levels and assess the outcomes according to the quartiles of the % change. We used the variables for this analysis same as the main analysis adding the duration of hospital stay. The results are mostly consistent with the main analysis. (Results, page 16)

 

Reviewer #2: In this paper, Kato et al. aimed at assessing the prognostic implication of a change in albumin during hospitalization for acute decompensated heart failure (ADHF) on prognosis.

The data of this study are derived from the Kyoto Congestive Heart Failure registry, a physician-initiated, prospective, observational, multicenter cohort study that enrolled consecutive patients hospitalized for ADHF October 2014 and March 2016.

This allowed the inclusion of an important number of patients (3160 patients).

The study is well written, and the results are interesting.

Nevertheless, I have some concerns that need to be addressed.

Response:

We thank the reviewers for careful assessment and positive comments.

1) The authors state that patients were divided into 2 groups according to the increase or the absence of increase in albumin levels before discharge.

How the increase in albumin level was defined. A simple increase in 1 unit of albumin was sufficient to define increase, or a stricter definition was used?

Response:

We thank the reviewer for the very important questions. The increase was defined as an increase in less than 1 unit. As the reviewer 1 pointed out, there may be an intra-individual variety; thus, we have added a sensitivity analysis using the % change and assess the outcomes according to the quartiles of the % change. The results were almost consistent with the main analysis. In the analysis, there seems to be a difference in outcomes between Q2 and Q3. 

However, the purpose of the present study is not seeking the threshold of albumin changes. Instead, we adopted the simpler definition i.e. increase or non-increase in the main analysis.

“Sensitivity analysis based on the percent change of albumin levels

We stratified patients into 4 groups based on the quartiles of the percent change of albumin levels: the lowest quartile (≤ -11.1%), the lower quartile (> -11.1% and ≤ -3.0%), the higher quartile (> -3.0% and ≤ 5.3%), and the highest quartile (> 5.3%) (S4 Table). The cumulative 1-year incidence of the primary outcome measure (a composite endpoint of all-cause death or hospitalization for HF) was significantly lower in the higher and the highest quartiles than that in the lowest quartile (S5 Figure). The lower risk of the higher and the highest percent change groups relative to the lowest percent change group was significant after adjustment (HR: 0.70, 95% CI: 0.58-0.83, P <0.0001) (S6 Table)”. (Results, page 16)

“We could not assess the intra-individual variability. In order not only to evaluate the prognostic impact of any increase in albumin but also to evaluate the increase that is above the intra-individual variability, we added the sensitivity analysis on the percent change of albumin levels. From a prognostic point of view, the cut-off point might exist around -3.0%; however, this value may be validated by further studies”. (Discussion, page 21, lines 12-16)

2) The paragraph on Ethics is too long. For the publication, I think it is necessary to]: indicate that “The investigation conformed with the principles outlined in the Declaration of Helsinki and approved by the ethical committees of the participating hospitals”. A detailed list of these hospitals and corresponding approbation numbers should be provided as supplementary material.

Response:

We have included the details of ethics in the supplementary material.

3) The paragraph “Definition” doesn’t identify the content of the paragraph itself. I suggest indicating the title of this paragraph as “Outcome”.

Response:

We thank the reviewer for the comments. I have changed the title of the paragraph.

4) The “detailed definition of patients characteristics should be inserted in the paragraph “Population”.

Response:

We have included the definition of patient’s characteristics should be inserted in the paragraph “Population”.

5) The authors should add some data about albumin assessment. The fact that they consider variation in albumin concentration during hospitalization is indicated in the introduction, but this should be clearly indicated in the method. They should indicate when albumin was assessed at admission and before discharge. Any increase in albumin level (even the smallest one) corresponded to albumin increase? This should be defined in detail.

Response:

We have included the data about albumin assessment including the timing of assessment in the method section. We clarify any increase in albumin level defined as an increase in albumin. 

“We compared albumin levels at admission/hospital presentation (anytime of the day) and at the nearest time to discharge in the morning. Any increase in albumin level corresponded to albumin increase.” (Methods, page 5, lines 19-21)

In order not only to evaluate the prognostic impact of any increase in albumin but also to evaluate the increase that is above the intra-individual variability, we added the sensitivity analysis on the percent change of albumin levels as mentioned above. 

6) Can the author indicate the cause of hospital admission?

Response:

We appreciate your comments. Unfortunately, the precipitating factors for the acute decompensation could not be clearly identified and not collected in the present study.

7) In table 1, the p-value for HFmEF and HFpEF is not indicated and should be added.

Response:

We have added the p-value in the table.

8) In table 1, the definition “Albumin, g/dl” should be replaced by “Increase in albumin levels”.

Response:

We have added the increase in albumin levels and the definition in Table 1.

9) In table 1, the p-value for the different quartiles of albumin is not indicated.

Response:

We have added the p-value in the table.

10) Were patients taking also non-MRA diuretics at discharge?

Response:

We have added the diuretics at discharge in the table 1. We thank the reviewer for the careful assessment.

11) Table 2 is difficult to read. I think the authors should simply indicate unadjusted an unadjusted HR in this table and maybe add the comparison of events among groups in Table 1. A similar comment can be applied to Table 3.

Response:

Thank you for your kind suggestion. In table 2 and 3, comparisons between the two groups was based on the time-considering method, such as proportional hazard method. In addition, it would be clearly presented how the unadjusted hazard ratio increases or decreases after the adjustment for confounders. Thus, we would like to keep the Table 2 and 3 in the present form in the revised manuscript.

12) From a clinical point of view, were all patients discharged in a condition of euvolemia? Did the study include patients admitted for cardiogenic shock?

Response:

We appreciate your very important comments. It is very difficult to say all patients discharge were in a condition of euvolemia. Instead, we have added the BNP levels at discharge and discussions about the volume status. We included all patients who had acute decompensated heart failure and underwent heart failure-specific treatment involving intravenous drugs within 24 hours of hospital presentation. Thus, patients with low blood pressure were included in the present study. We included the systolic blood pressure <90mmHg into the adjusting variables.

“In addition, patients with no increase in albumin levels had a higher BNP levels at discharge, although the difference was not large. From a clinical point of view, the assessment of intravascular volume status is crucial when we interpret the change of albumin levels.”(Page 22, lines 8-11)

13) In the conclusion the authors state that “Independent of baseline albumin levels, an increase of albumin was associated with a lower 1-year risk for the composite of all-cause death or HF hospitalization in patients hospitalized for ADHF”. P-value for the adjusted risk of HF is >0.05, which means that an increase in albumin level is not associated with a significant reduction in HF hospitalization. The statement should be corrected and these results should put in perspective in the discussion.

Response:

We thank the reviewer for the valuable comments. Primary outcome is a composite outcome but the differences in HF hospitalization were not statistically significant. As the reviewer pointed out, we have specified the outcomes to avoid the unclearness in the conclusion section, and added the perspectives in the discussion section.

“whether the increase of albumin can be associated with a lower risk for HF hospitalization should be determined by further studies.” (Discussion, page 22, lines 16-18)

“Conclusion Independent of baseline albumin levels, an increase of albumin was associated with a lower 1-year risk for all-cause death in patients hospitalized for ADHF.” (Page 23, lines 8-10)

14) I think that all the survival curves should indicate the adjusted comparison of survival/hospitalization curve and not unadjusted data. Otherwise, the results are largely misleading.

Response: 

We appreciate your comments. We described the cumulative incidence of each group in the text and main figures consistently. We totally agree on the reviewer’s suggestion; thus, we have added the adjusting survival curves in the supplementary figures. If the editors and reviewers strongly suggests that this result should be included in the main figure, we will be glad to add the data in the main figures. 

15) Were the authors able to identify parameters associated with the increase in albumin levels during hospitalization?

Response: 

We appreciate your comments. We have included the multivariate analysis regarding the baseline factors associated with the increase in albumin levels.

“To analyze the baseline factors associated with high LVMI, we used a multivariable logistic regression model involving the following potential independent clinically relevant variable. The adjusted odds ratios and 95% CIs were calculated.” (Methods, page 9, lines 8-10)

“Baseline factors associated with the increase in albumin levels

According to the multivariable logistic regression analysis, the albumin levels at admission < 3 g/dL and anemia were independent positive factors associated with the increase in albumin levels, while age ≥80 years, eGFR<30 ml/min/1.73m2, body mass index<22kg/m2, NYHA III/IV at presentation, and the history of diabetes mellitus were independent negative factors associated with the increase in albumin levels (Table 4).” (Results, page 19-20)

---

## [Decision Letter · Decision Letter 1]

27 Nov 2020

Association of an increase in serum albumin levels with positive 1-year outcomes in acute decompensated heart failure: A cohort study

PONE-D-20-30330R1

Dear Dr. Kato,

We’re pleased to inform you that your manuscript has been judged scientifically suitable for publication and will be formally accepted for publication once it meets all outstanding technical requirements.

Kind regards,

Vincenzo Lionetti, M.D., PhD

Academic Editor

PLOS ONE

Additional Editor Comments (optional):

Reviewers' comments:

Reviewer's Responses to Questions

**Comments to the Author**

1. If the authors have adequately addressed your comments raised in a previous round of review and you feel that this manuscript is now acceptable for publication, you may indicate that here to bypass the “Comments to the Author” section, enter your conflict of interest statement in the “Confidential to Editor” section, and submit your "Accept" recommendation.

Reviewer #1: All comments have been addressed

Reviewer #2: All comments have been addressed

2. Is the manuscript technically sound, and do the data support the conclusions?

Reviewer #1: Yes

Reviewer #2: Yes

3. Has the statistical analysis been performed appropriately and rigorously? 

Reviewer #1: Yes

Reviewer #2: Yes

4. Have the authors made all data underlying the findings in their manuscript fully available?

Reviewer #1: Yes

Reviewer #2: Yes

5. Is the manuscript presented in an intelligible fashion and written in standard English?

Reviewer #1: Yes

Reviewer #2: Yes

6. Review Comments to the Author

Reviewer #1: The Authors have modified their manuscript according to my suggestions. I have no further comments.

Reviewer #2: The authors dealt with al the comments raised by the reviewed. They added supplementary statistical analysis. The current version of the manuscript is significantly improved and I do not propose further modifications. I think that the current version of the article is suitable for publication.

7. PLOS authors have the option to publish the peer review history of their article (what does this mean?). If published, this will include your full peer review and any attached files.

Reviewer #1: No

Reviewer #2: No

---

## [Editor Report · Acceptance letter]

11 Dec 2020

PONE-D-20-30330R1 

Association of an increase in serum albumin levels with positive 1-year outcomes in acute decompensated heart failure: A cohort study  

Dear Dr. Kato:

I'm pleased to inform you that your manuscript has been deemed suitable for publication in PLOS ONE. Congratulations! Your manuscript is now with our production department. 

Kind regards, 

on behalf of

Prof. Vincenzo Lionetti 

Academic Editor

PLOS ONE